# Modified Uni-Traveling-Carrier Photodetector with Its Optimized Cliff Layer

**DOI:** 10.3390/s24072020

**Published:** 2024-03-22

**Authors:** Xiaowen Dong, Kai Liu

**Affiliations:** State Key Laboratory of Information Photonics and Optical Communications, School of Electrical and Electronic Engineering, Beijing University of Posts and Telecommunications, Beijing 100876, China; dxw_1998@bupt.edu.cn

**Keywords:** electric-field pre-distort, saturation current density, optimized cliff layer

## Abstract

We have designed the MUTC-PD with an optimized thickness of cliff layer to pre-distort the electric field at the front side of the collection layer. With the optimized MUTC-PD design, the collapse of the electric field will be greatly suppressed, and the electrons in its collection layer will gradually reach their peak velocity with the growing incident light power. Moreover, as the incident light intensity increases, the differential capacitance also declines, thus the total bandwidth grows. It will make the MUTC-PD achieve high-speed and high-power response performance simultaneously. Based on simulation, for 16μm MUTC-PD with a 70 nm cliff layer, the maximum 3 dB bandwidth at −5 V is 137 GHz, compared with 64 GHz for the MUTC-PD with a 30 nm cliff layer. The saturation RF output power is 27.4 dBm at 60 GHz.

## 1. Introduction

Terahertz technology is attracting more and more interest in recent years because it is widely used in the fields of wireless communications [1,2,3], biosensors [4,5,6], imaging [7,8], and manufacturing control. Semiconductor optoelectronic devices can facilitate THz-wave generation and detection [9,10,11,12,13,14], which are two basic parts for terahertz sensing applications. Among them, uni-traveling-carrier photodiode (UTC-PD), especially modified UTC-PD (MUTC-PD), shows the widest dynamic range and the best high-speed, high-power performance, compared with PIN-PD and APD [15,16,17,18,19]. However, the high-speed response of UTC-PD (or MUTC-PD) is mutually constrained by its RF-output performance according to the author of [20]. The bandwidth of UTC-PD (or MUTC-PD) initially increases with the increase in current density, but sharply decreases when the current density exceeds a certain limit. The limit value is set to be PD’s saturation output current density, which is called $J_{max}$. The $J_{max}$ can be improved in two ways, one is to raise the collection layer’s doping concentration, and the other is to enhance electrons’ effective transit speed in the collection layer.

It is proposed that increasing the n-type doping concentration in UTC-PD’s collection layer within a certain range can relieve the space charge effect caused by saturation output current and enhance the saturation performance of the device [21]. However, excessively high doping concentrations will deteriorate PD’s breakdown characteristics. Thus, researchers mainly utilized the electron’s transient overshoot characteristic to increase electron’s equivalent drift velocity in depleted layer and improve the saturation current density [11,22,23,24]. The near-ballistic UTC-PD achieves a saturation output current of 30 mA by adjusting the electric field intensity to make electrons in the collection layer transport at an overshoot velocity [22]. It is demonstrated that the electron drift velocity increases from 9 × 10^6^ cm/s to 4 × 10^7^ cm/s when the photocurrent increases from 1 mA to 20 mA [23]. In addition to the influence of photocurrent on carriers’ transport and velocity, changes in bias voltage can also adjust the distribution and transport of carriers inside the device, so the current output can be precisely controlled [24]. However, space-charge screening (SCS) is also a major issue, limiting the saturation current of a PD. The SCS effect slows down the electron drift process in the collection layer, resulting in speed degradation and limiting the maximum output power. The N-UTC-PD is designed to hold a significant electron potential drop at the junction of the absorption layer and collection layer [11]. Thus, the SCS effect is effectively suppressed. Researchers proposed a dual-drift layer UTC-PD (DDL UTC-PD) for simultaneously acquiring better saturation characterization and high-speed performance [25]. Also, through applying electron velocity overshoot in the absorption layer, the 4.5 μm diameter mesa-structure MUTC-PD reported in [18] has demonstrated a 3-dB bandwidth of over 150 GHz with a responsivity of 0.165 A/W. However, the effective overshoot distance can only maintain a very short distance, usually less than 500 nm, and can be used only for PD with a small sensing area to compensate for the parasitic capacitance effect. It has been investigated that the 20 μm diameter MUTC-PD with a 300 nm thick collection layer only achieves a bandwidth of 40 GHz after epitaxy optimization [26].

For thick collection layers longer than 500 nm, the electron’s transient overshoot characteristic will lose its advantage, and PD’s saturation output current density $J_{max}$ cannot acquire great improvement. Under such conditions, the electron’s transit peak velocity characteristic makes up for this drawback [27]. According to the electric field-velocity relationship of the electron’s transit performance in InP, under certain low electric field intensity regions, the electron can travel at its peak velocity, which is times of its saturation velocity. In this way, the UTC-PD can have a thicker collection layer without increasing the electron’s transit time. A thicker collection layer can bear a lager optical sensing window diameter without deteriorating the UTC-PD’s parasitic capacitance effect, which means the UTC-PD can achieve a higher saturation current with the same $J_{max}$ condition. However, another drawback exists under high input light intensity condition, which will deteriorate the UTC-PD’s performance. That is the electron accumulation at the PN heterojunction, where the cliff layer is located. The accumulation will be high and result in a decrease in the PD’s saturation performance. Thus, the effects of the cliff layer’s doping concentration and thickness changes on the UTC-PD’s bandwidth under fixed working conditions are investigated [28]. Researchers also studied the impact of changes in cliff layer position on electric field distribution in the absorption layer and the collection layer. After optimization, the MUTC-PD with a 22 μm diameter has an output photocurrent of 99 mA at 28 GHz and a saturation RF output of 20.1 dBm [29].

In this paper, the above working scheme will be analyzed in detail. And we compare the electric field, electron velocity, and electron concentration distributions for the MUTC-PD with different thicknesses of cliff layer. For the MUTC-PD with a 70 nm cliff layer, the voltage dropped on its cliff layer will pre-distort the electric field in its collection layer and make the collection layer not totally depleted. Under working conditions, with increasing the incident light power, the MUTC-PD’s collection layer will be depleted gradually due to Kirk’s effect. Thus, the electron transit performance within the MUTC-PD can be optimized for high power and high-speed performance. With an optimized design, the electron velocity in MUTC-PD’s collection layer will keep rising, and its depletion region will gradually expand with the increased incident light power so that its electron transit velocity can maintain its peak velocity under higher incident optical power. For comparison, in the MUTC-PD with a 30 nm cliff layer, its collection layer will be depleted, and its electron transit velocity will reach saturation velocity at a lower injected light intensity, resulting in a deteriorated saturation performance. We analyze the scheme for their different trends in total bandwidth as light intensity grows. The MUTC-PD with a 70 nm cliff layer shows a maximum bandwidth of 137 GHz and a saturation RF-output power of 27.4 dBm. As compared, the MUTC-PD with a 30 nm cliff layer only has 60 GHz of maximum bandwidth and 20.8 dBm of saturation RF-output power.

## 2. Device Structure Design and Simulation Analysis

### 2.1. Simulation Method

The simulation software used in this study is Silvaco, Santa Clara, CA, USA, Deckbuild Atlas 4.6.2.R [30]. The simulation analysis of semiconductor device performance is conducted by solving the self-consistent solution of the basic equation used to describe the operation of semiconductor devices. The basic equations contain Poisson’s equation, carrier continuity equation, and current density equations. And the physic models consisting of parallel electric field mobility and negative differential mobility models can reflect the properties that, as the electric field increases, the velocity of carriers first increases to peak value and then tends to saturation due to carriers’ scattering. The material parameters of InGaAs and InP used in this simulation are the same as those given in [31].

### 2.2. Epitaxy Structure Design

In InP material, electron velocity can reach a peak value at about 11 kV/cm field intensity, which is called the electron peak velocity. As the electric field intensity increases, the electron velocity gradually decreases until it reaches saturation. Compared with the electron overshoot effect, the electron peak velocity effect can sustain a very long distance. According to the saturation current density equation, as follows:(1)$$J_{max}=qv_{sclimit}N_{DC}$$
where $v_{sclimit}$ is the saturation electron velocity, $N_{DC}$ is the doped concentration in PD’s depletion layer. The MUTC-PD can generate much higher output current by elevating the electron velocity into peak velocity.

The epitaxy layers of the MUTC-PD are shown in Figure 1. According to the above analysis, the InP collection layer is designed to be as thick as 1000 nm for two purposes: one is to make the electric field intensity in the collection layer lower than the intensity at the electron velocity saturation point, so that the electron velocity in the collection layer has a certain range that increases with growing light intensity. The other purpose is to reduce the UTC-PD’s parasitic capacitance for 16~20 μm of large diameter. The resulting kirk effect brings a further decrease in capacitance. However, even with a wide depletion layer, achieving the electron peak velocity also requires the photodiode to have a specific cliff layer thickness, which greatly affects MUTC-PD’s working characteristics. Therefore, the simulation includes an analysis of cliff layer thickness ranging from 20 to 70 nm, especially analyzing and comparing the effects of 30 nm (thin cliff layer) and 70 nm (thick cliff layer) on the operating characteristics of the MUTC-PD. The MUTC-PD also contains a 200 nm thick p-doped InGaAs absorption region and a 200 nm thick depleted InGaAs absorption region to balance the electrons’ diffusion and holes’ drift. The responsivity is 0.39 A/W.

### 2.3. Theoretical Analysis and Characteristics Comparison

We focus on the simulation values at the interface of the absorption layer and space layer (line A), and the value at the cliff layer (line B), which are marked in Figure 1. The difference values of electron velocity and electron concentration between the two lines (A and B), which represent the degree of electron accumulation with increasing light intensity, are depicted in Figure 2a,b. For the cliff layer’s thickness of 20 to 40 nm, the electron velocity difference shown in Figure 2a gradually decreases, which indicates that the electrons emitted from the absorption layer generate an electron jam at the heterojunction barrier as light intensity grows. For a thickness of 50 to 70 nm, electron transit maintains smoothness with the increasing light intensity. Correspondingly, with increasing light intensity, the difference value of electron concentration hardly grows for PD with a cliff layer’s thickness of 50 to 70 nm, but has a significant increase for PD with a cliff layer’s thickness of 20 to 40 nm, as shown in Figure 2b.

We further simulate the MUTC-PDs with 30 nm and 70 nm cliff layers under different incident light intensities to deeply analyze their internal differences. The bias voltage is both −5 V. For the MUTC-PD with a 30 nm cliff layer, the potential drop is comparatively low in the cliff layer and high in the collection layer. Thus, its collection layer is completely depleted under 1 × 10^4^ W/cm^2^, as shown in Figure 3a. As the light intensity increases, the slope of the electric field intensity in the collection layer continuously changes, but the depletion width almost no longer increases, thus the PD’s capacitance is basically not reduced. In contrast, as Figure 3b shows, for the MUTC-PD with a 70 nm cliff layer, the electric field intensity in its cliff layer is significantly higher than that of 30 nm under the same light intensity. Also, the collection layer gradually becomes depleted with the increasing light intensity, so that the capacitance continues to decline with the growing light intensity. And the electric field intensity at the cliff layer side in the collection layer reaches zero under 1.6 × 10^5^ W/cm^2^. This indicates that a field collapse occurs there, and the MUTC-PD reaches saturation. Besides, as shown in Figure 4a, for MUTC-PD with a 30 nm cliff layer, the electron velocity has a slight change under 1 × 10^4^ W/cm^2^~5 × 10^4^ W/cm^2^ and then reaches the electron saturation velocity value in the entire collection layer and keeps it constant above 5 × 10^4^ W/cm^2^. Figure 5a vividly illustrates the overall trend of electron velocity changes in the collection layer for the PD with a 30 nm cliff layer. As the injection layer intensity increases, the average electron velocity in the collection layer gradually declines to the saturation value. After reaching the saturation value, the increase in electric field intensity caused by the increase in light intensity will not further improve the electron velocity. As a comparison, Figure 4b shows that the electron velocity at the n-contact side of the collection layer changes greatly with the growing light intensity under 8 × 10^4^ W/cm^2^, and the electron velocity at the cliff layer side of the collection layer gradually uplifts with the growing light intensity over 8 × 10^4^ W/cm^2^. Corresponding to the point where the electric field collapses under 1.6 × 10^5^ W/cm^2^, as shown in Figure 3b, the electron velocity descends to near zero at that point. Electrons accumulate severely, so the MUTC-PD saturates. Figure 5b reveals the overall trend of electron velocity changes in the collection layer for the PD with a 70 nm-cliff layer. As the injection layer intensity increases, the average electron velocity in the collection layer will gradually climb to its peak value until the space charge effect appears. The different rules for the two MUTC-PDs will lead to differences in PD saturation and bandwidth performance.

For the PD with the cliff layer’s thickness of 30 nm, in order to exclude the impact of the high bias voltage applied on the variation of electric field intensity in the collection layer, we also simulate the PD’s electric field intensity and electron concentration distributions under −4 V. The collection layer for the PD with a 30 nm-cliff layer is undepleted at 1 × 10^4^ W/cm^2^~3 × 10^4^ W/cm^2^, as shown in Figure 6a. According to Figure 6b, the spike of electron velocity in the collection layer gradually moves to the side near the n-contact layer, and after 7 × 10^4^ W/cm^2^, the electron velocity in the collection layer has slight improvement at the cliff layer side. Although the changing trend seems similar to the PD with a cliff layer’s thickness of 70 nm, the accumulation of electrons inside them is substantially different. Figure 7a,b represents the changes with different incident light intensities in electron concentration for PD with a 30 nm cliff layer under −4 V and PD with a 70 nm cliff layer under −5 V. The electron concentration at the barrier of heterojunction of the former is one order of magnitude larger than that of the latter. For further analysis, the transit-time-decided bandwidth, total bandwidth, and PD’s capacitance for the PD with the cliff layer’s thickness of 30 nm are simulated, as shown in Figure 8. It can be summarized that, although the capacitance declines linearly at the beginning, the total bandwidth is determined by the continuously decreasing transit-time-decided bandwidth. When the light intensity increases to 6 × 10^4^ W/cm^2^, the capacitance also increases because the thickness of the depletion region no longer expands and electron accumulation causes the growth of electrons’ total transit time. Therefore, PD’s bandwidth shows a downward trend.

### 2.4. Performance Comparison and Analysis

The dark current profiles are shown in Figure 9. The dark current of MUTC-PD with cliff layer thicknesses of 30 nm and 70 nm is both below 1 nA. This level of dark current will not affect the sensitivity of photodiodes for receiving optical signals. We compare the changing rules of high-speed performance and linearity performance between these two types of photodiodes in terms of changing light intensities under −5 V. Figure 10a shows the transit-time-decided bandwidth, total bandwidth, and capacitance of PD with a 30 nm cliff layer at different incident light intensities. The $f_{RC}$ is MUTC-PD’s RC-time-decided bandwidth, which is calculated using the formula $f_{RC}=\frac{1}{2\pi RC}$. The $f_{t}$ is MUTC-PD’s transit-time-decided bandwidth obtained from simulation. The bias voltage is −5 V. Although the capacitance keeps declining under 3 × 10^4^ W/cm^2^, the total bandwidth falls continuously, following the falling transit-time-decided bandwidth due to the $f_{RC}>f_{t}$. When the light intensity is greater than 3 × 10^4^ W/cm^2^, both the transit-time-decided bandwidth and the RC-time-limited bandwidth fall, so the total bandwidth shows a constant downward trend. The highest bandwidth is only 64 GHz at a light intensity of 1 × 10^4^ W/cm^2^. Figure 10a shows the bandwidth and capacitance data of PD with a 70 nm cliff layer. The bias voltage is also −5 V. When the light intensity is below 8 × 10^4^ W/cm^2^, the transit-time-decided bandwidth $f_{t}$ is greater than the RC-time-limited bandwidth $f_{RC}$, so the trend of the total bandwidth with the growing light intensity is consistent with the trend of the capacitance changes. When the light intensity is beyond 8 × 10^4^ W/cm^2^, both the transit-time-decided bandwidth $f_{t}$ and the RC-time-limited bandwidth $f_{RC}$ increase with the growing light intensity. Therefore, the total bandwidth rises at a greater slope compared with its slope below 8 × 10^4^ W/cm^2^. The better frequency response at a higher incident light power depends not only on the electron velocity gradually approaching the peak velocity in the collection layer. It can also be explained using Figure 3b. Due to the high divided voltage in MUTC-PD’s cliff layer, the electric field is pre-distorted on the cliff layer side in the collection layer. Thus, the collapse of the electric field at higher incident light power is suppressed. The highest bandwidth is achieved at a light intensity of 1.3 × 10^5^ W/cm^2^. The frequency response at 1.3 × 10^5^ W/cm^2^ is depicted in Figure 11. The transit-time-decided bandwidth and total bandwidth are 197 GHz and 137 GHz, respectively. Figure 12a and Figure 12b, respectively, show the saturation performance of PD with a 30 nm and 70 nm cliff layers. The RF frequency is 60 GHz, and the bias voltage is −5 V. The ideal power (black line in Figure 12a) is the theoretical RF output power when the photodiode is modeled as an ideal current source. The PD with a 70 nm cliff layer shows a broader linear dynamic range. The maximum RF output power of the PD with a 70 nm cliff layer is 27.4 dBm, as shown in Figure 12a, compared with 20.8 dBm for the PD with a 70 nm cliff layer, as shown in Figure 12b.

## 3. Conclusions

In this paper, we have optimized the MUTC-PD’s cliff layer to improve its high-speed and saturation performance. We have compared the MUTC-PDs with thin cliff layer (30 nm) and thick cliff layer (70 nm). Through increasing the cliff layer’s thickness, the potential drop in it is comparatively higher, thus the electric field in the collection layer is pre-distorted. The collapse of the electric field at higher incident light power is suppressed. For the two types of MUTC-PD, simulation results show that the main factors affecting the variation trend of PD’s total bandwidth at different incident light intensities are the lower ones between transit-time-decided bandwidth and RC-time-limited bandwidth. For a 16μm MUTC-PD with a 70 nm cliff layer, the maximum 3 dB bandwidth is 137 GHz at a bias voltage of −5 V, and the maximum RF output power at 60 GHz is 27.4 dBm.

## Figures and Tables

**Figure 1 sensors-24-02020-f001:**
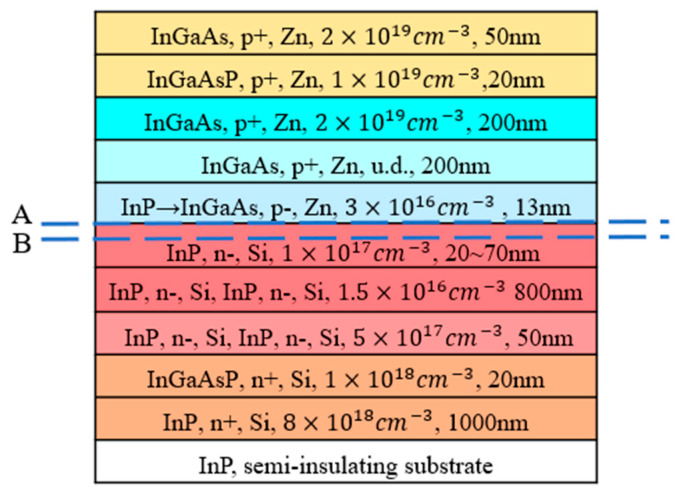
Epitaxy layer of the MUTC-PD, line A: at the interface of the absorption layer and space layer, line B: at the cliff layer.

**Figure 2 sensors-24-02020-f002:**
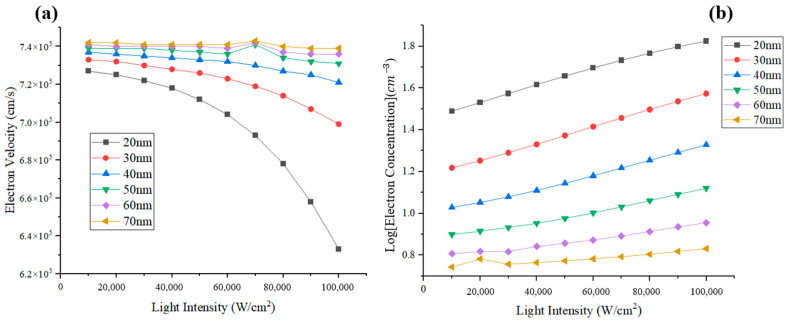
The difference values of (**a**) electron velocity and (**b**) electron concentration between the two points for the MUTC-PD with a cliff layer thickness of 20~70 nm.

**Figure 3 sensors-24-02020-f003:**
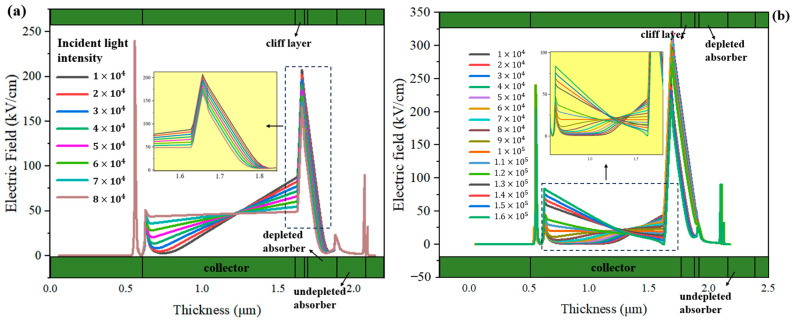
The change in electric field distributions with different incident light intensities of (**a**) the MUTC-PD with a 30 nm cliff layer and (**b**) the MUTC-PD with a 70 nm cliff layer.

**Figure 4 sensors-24-02020-f004:**
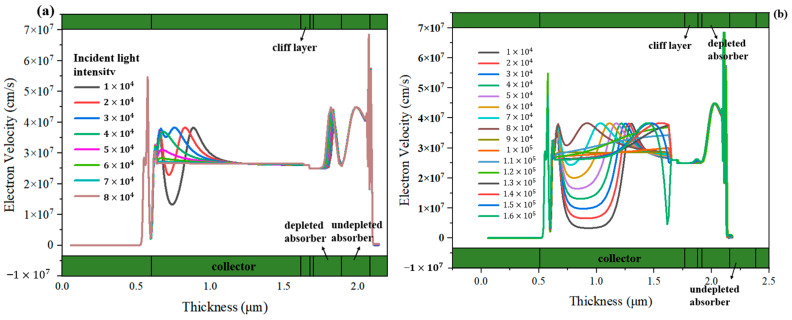
The change in electron velocity distributions with different incident light intensities of (**a**) the MUTC-PD with a 30 nm cliff layer and (**b**) the MUTC-PD with a 70 nm cliff layer.

**Figure 5 sensors-24-02020-f005:**
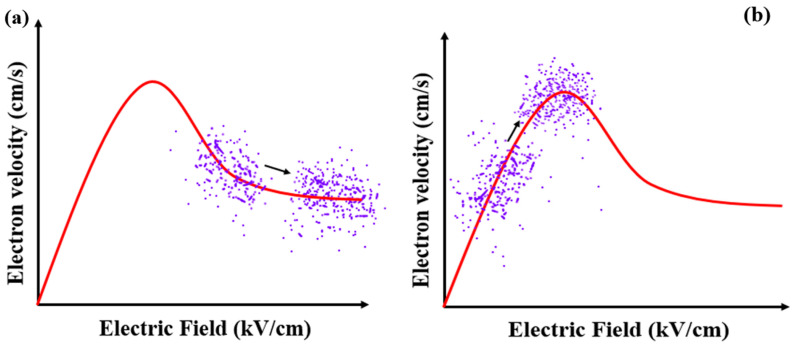
The overall trend of electron velocity changes in the collection layer for (**a**) the MUTC-PD with a 30 nm cliff layer and (**b**) the MUTC-PD with a 70 nm cliff layer. (The dots represent the velocity distribution of electrons).

**Figure 6 sensors-24-02020-f006:**
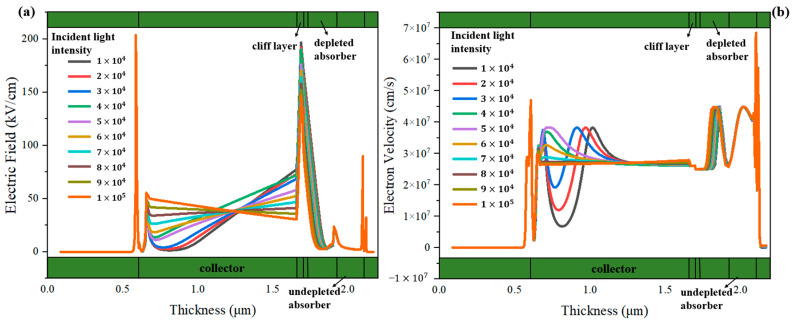
The 30 nm-cliff layer MUTC-PD’s (**a**) electric field intensity and (**b**) electron velocity distributions under −4 V.

**Figure 7 sensors-24-02020-f007:**
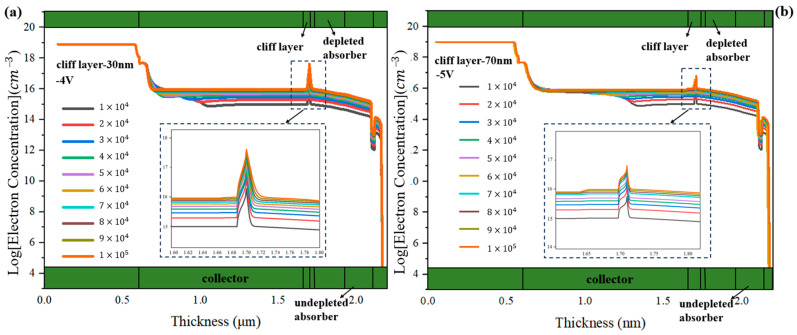
The changes with different incident light intensities in electron concentration for (**a**) PD with a 30 nm cliff layer under −4 V and (**b**) with a 70 nm cliff layer under −5 V.

**Figure 8 sensors-24-02020-f008:**
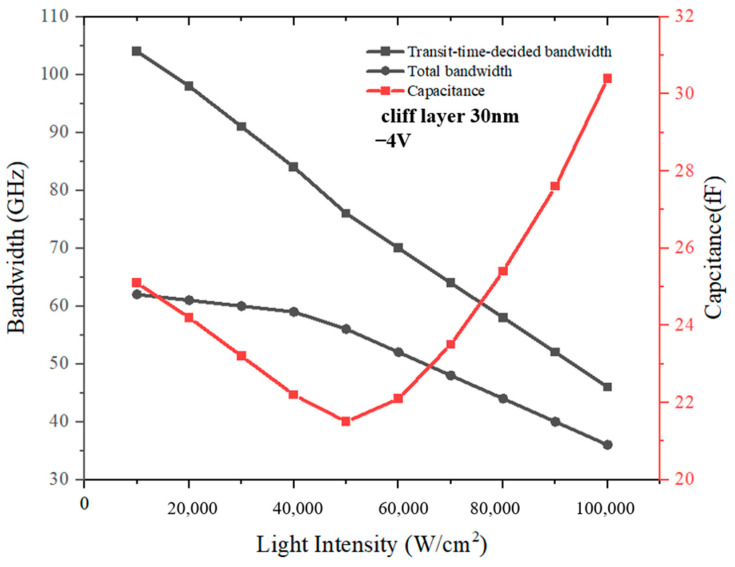
The frequency response and capacitance with different light intensities for the MUTC-PD with a 30 nm cliff layer under −4 V.

**Figure 9 sensors-24-02020-f009:**
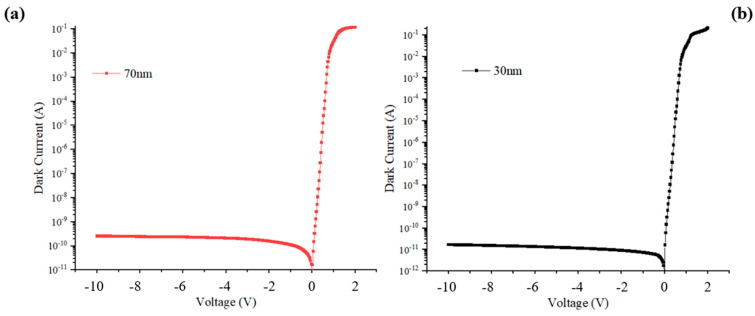
The dark currents for (**a**) the MUTC-PD with a 70 nm cliff layer and (**b**) the MUTC-PD with a 30 nm cliff layer under −5 V.

**Figure 10 sensors-24-02020-f010:**
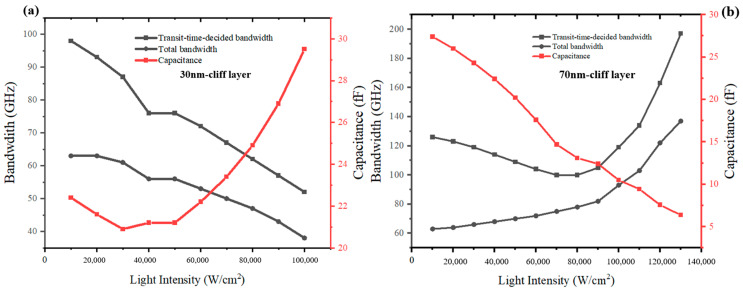
The frequency response and capacitance with different light intensities for (**a**) the MUTC-PD with a 30 nm cliff layer and (**b**) the MUTC-PD with a 70 nm cliff layer under −5 V.

**Figure 11 sensors-24-02020-f011:**
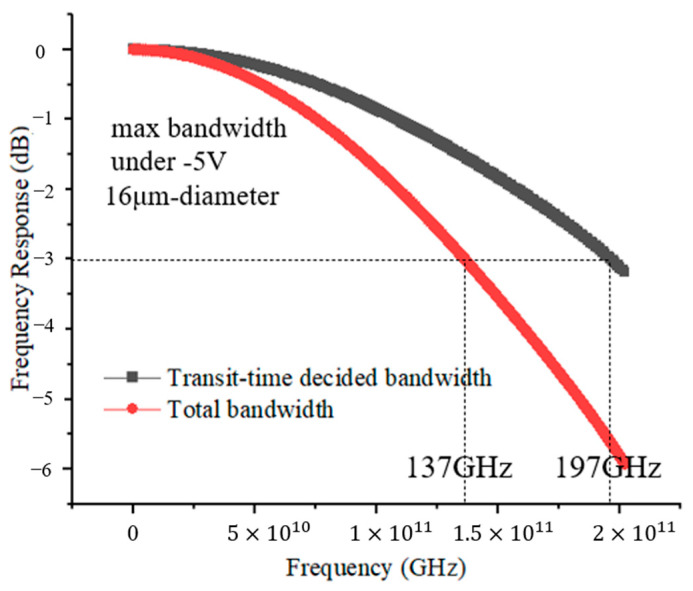
The frequency response of MUTC-PD with a 70 nm cliff layer at 1.3 × 10^5^ W/cm^2^.

**Figure 12 sensors-24-02020-f012:**
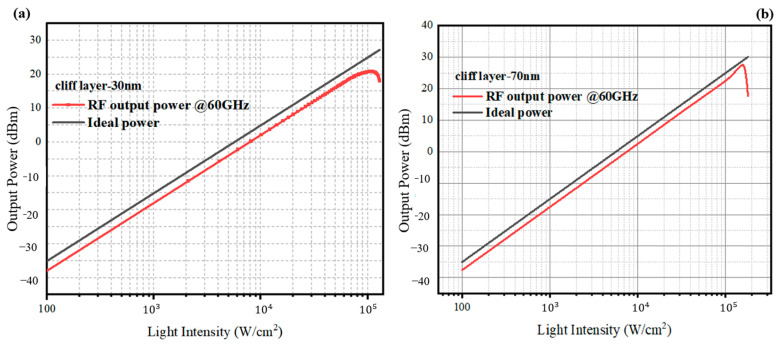
The RF output power for (**a**) the MUTC-PD with a 30 nm cliff layer and (**b**) the MUTC-PD with a 70 nm cliff layer under −5 V.

## Data Availability

Data are contained within the article.

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
