# Peer review of "Modified Uni-Traveling-Carrier Photodetector with Its Optimized Cliff Layer"

_sensors, 2024, doi:10.3390/s24072020_

Round 1

Reviewer 1 Report

Comments and Suggestions for Authors

Dear Editor,

This manuscript discusses the design of a detector that has a good performance compared to the APD structure. Various challenges need to be answered. Therefore, I cannot recommend the article for publication now; the following revisions are necessary.

1. High input light power can cause nonlinear effects. Are these effects considered?

2. In Figure 5, why is the speed of electrons decreased after overshoot?

3. Dark current and GBP have not been calculated to be investigated.

4. The role of the cliff layer in this research should be compared with the charge layer in SACM-APD.

5. There are various ideas in the following articles that should be compared with this research and discussed in the introduction section.

Journal of Nanoelectronics and Optoelectronics 17.3 (2022): 495-504, Optical and Quantum Electronics 54.3 (2022): 171

6. The time response of the device to the square light pulse should be checked to determine the rise time and fall time.

7. One of the important features of optical detectors is the access noise factor, which is not investigated in this manuscript.

Kind regards

Reviewer 2 Report

Comments and Suggestions for Authors

The authors present an improvement of an UTC photodiode by adjusting the cliff layer thickness. While their work is intereisting the submitted paper lacks in-depth analysis and comparison with other works improving the performance of such photodiodes. Additionally it is important that the authors perform some more simulations to underline the results. It would be good, if they could do the simulation of the 70 nm variant until a point where this photodiode reaches saturation. How does the velocity of the 70 nm cliff layer version looks like in dependence of the electric field ( fig. 5). Please also rework the description of fig. 4 b. An improvement of the overall presentation is very important, since some figures next to each other have different scales making a comparison almost impossible. Because of the these reasons I have to recommend a major revision.

Round 2

Reviewer 1 Report

Comments and Suggestions for Authors

Dear Editor,

The manuscript has been revised based on the comments so I think the present version meets the acceptance level.

Kind regards,

Author Response

Thank you for the positive comments.

Reviewer 2 Report

Comments and Suggestions for Authors

Thank you very much for the correction of the manuscript. I think these improve the quality of the paper already very much, but I am still missing the point where the PD with 70 nm cliff layer is in saturation. Please add higher optical input powers to figure 12. In fig. 12 b is also the mistake that the legend of the colors is wrong. Please correct it as in fig. 12a. The saturation of the PD is important to understand, because the ideal and the PD line are almost crossing each other, which is impossible.
